# Preventing cardiotoxicity in patients with breast cancer and lymphoma: protocol for a multicentre randomised controlled trial (PROACT)

Rebecca H Maier [1,2,3] Chris Plummer,[4] Adetayo S Kasim [5]
Nasima Akhter [5] Emmanuel Ogundimu,[6] Jamie Maddox [7] Janine Graham,[8]
Michael Stewart,[9] Andrew Wardley,[10] Sophie Haney,[11] Sharareh Vahabi,[9]
Helen Oxenham,[12] Alison Humphreys,[8] Nicola Cresti [13] Mark Verrill,[13]
Richard Graham,[9] Lisa Chang,[2] Helen C Hancock [1,3] David Austin [2,3,9]

For numbered affiliations see end of article.

**Correspondence to**
Rebecca H Maier;
Rebecca.Maier@newcastle.ac.uk

## ABSTRACT

**Introduction** Anthracyclines are included in chemotherapy regimens to treat several different types of cancer and are extremely effective. However, it is recognised that a significant side effect is cardiotoxicity; anthracyclines can cause irreversible damage to cardiac cells and ultimately impaired cardiac function and heart failure, which may only be evident years after exposure. The PROACT trial will establish the effectiveness of the ACE inhibitor enalapril maleate (enalapril) in preventing cardiotoxicity in patients with breast cancer and non-Hodgkin's lymphoma (NHL) receiving anthracycline-based chemotherapy.

**Methods and analysis** PROACT is a prospective, randomised, open-label, blinded end-point, superiority trial which will recruit adult patients being treated for breast cancer and NHL at NHS hospitals throughout England. The trial aims to recruit 106 participants, who will be randomised to standard care (high-dose anthracycline-based chemotherapy) plus enalapril (intervention) or standard care alone (control). Patients randomised to the intervention arm will receive enalapril (starting at 2.5 mg two times per day and titrating up to a maximum dose of 10 mg two times per day), commencing treatment at least 2 days prior to starting chemotherapy and finishing 3 weeks after their last anthracycline dose. The primary outcome is the presence or absence of cardiac troponin T release at any time during anthracycline treatment, and 1 month after the last dose of anthracycline. Secondary outcomes will focus on cardiac function measured using echocardiogram assessment, adherence to enalapril and side effects.

**Ethics and dissemination** A favourable opinion was given following research ethics committee review by West Midlands—Edgbaston REC, Ref: 17/WM/0248. Trial findings will be disseminated through engagement with patients, the oncology and cardiology communities, NHS management and commissioning groups and through peer-reviewed publication.

**Trial registration number** NCT03265574.

## STRENGTHS AND LIMITATIONS OF THIS STUDY

⇒ Strong and ongoing patient and public involvement and clinical consensus informed the robust randomised trial design, with 80% power.
⇒ Multicentre UK trial with blinding of the primary outcome and the echo core laboratory reduces bias and will enable robust findings.
⇒ The intervention is low cost, and with decades of evidence indicating its safety; if findings from PROACT demonstrate effectiveness, its translation into patient benefit can be realised through rapid adoption into usual care.
⇒ Findings will have implications for multiple cancers across adult and paediatric settings where patients receive high-dose anthracycline chemotherapy.
⇒ Two patient groups (breast cancer and lymphoma) may make interpretation of results difficult, if differences between the groups are found.

## INTRODUCTION

Breast cancer is the most common malignancy among women worldwide, with over 55 000 new UK cases every year.[1 2] Lymphoma is the most common haematological malignancy and the sixth most common cancer in the UK, with non-Hodgkin's lymphoma (NHL) affecting 14 000 patients per year.[3] Treatment for both conditions includes anthracycline chemotherapy (epirubicin or doxorubicin); however, anthracyclines can cause immediate, irreversible damage to cardiac cells and ultimately impaired cardiac function and heart failure which may only be evident years after exposure.[4] The long-term incidence is approximately 5% and may be higher in older patients.[5–7] Asymptomatic left ventricular systolic dysfunction (LVSD) is more common, evident in 9.7% of patients with breast cancer within 12 months.[8] LVSD affects cardiac prognosis and limits subsequent cancer treatment options, particularly those that target the HER2 pathway such

BMJ

as trastuzumab (Herceptin).[8] [9] In lymphoma, due to significant improvements in survival rates, there is also greater prevalence of anthracycline-induced cardiotoxicity in survivors.[10] Waiting until anthracycline-induced heart failure becomes clinically evident is ill-advised, with response to treatment poor and survival at 2 years of 40%; considerably worse than other causes of heart failure.[11]

The burden of cardiotoxicity is particularly important in both patient groups, where 78% (breast) and 63% (NHL) will be alive 10 years following their cancer diagnosis.[1] [12] Preventing cardiac damage would offer clear benefits to patients and substantial costs savings to the NHS.

## Current evidence supporting the rationale for the trial

Anthracycline toxicity is thought to be due to the generation of reactive oxygen species causing cell death.[13] [14] ACE inhibitor (ACEI) promote nitric oxide and inhibit the production of angiotensin II, which in turn reduces $NAD(P)H$ oxidase responsible for superoxide formation. ACEI and angiotensin inhibition are protective of apoptosis in vascular endothelium and cardiac cells, prevent anthracycline cardiotoxicity in animal models and reduce oxidative stress.[15–19]

ACEIs are of proven benefit in cardiovascular medicine, and are widely used, well tolerated and inexpensive (cost £2 per patient per month). Enalapril is the reference standard ACEI and has been extensively studied.[20] [21]

Treatment to prevent cardiotoxicity is attractive and feasible, but is unproven. A Cochrane systematic review was unable to make definitive conclusions regarding the effectiveness of previously studied cardioprotective agents; no studies of ACEI were included.[22] A meta-review and further systematic review also found a lack of evidence to guide decision-making.[23] [24] A European Society of Cardiology (ESC) position paper and the recently published ESC guideline on cardio-oncology also draw the same conclusions on preventative therapy.[25] [26]

An Italian study assessed enalapril treatment in patients with a positive troponin during chemotherapy (not all were anthracycline-based regimens). By the second month, 41% of patients in the control group (n=58) had persistent troponin elevation, compared with 4% in the enalapril group (n=56). Based on echocardiography, no cardiotoxicity in the enalapril group was observed; cardiotoxicity was identified in 43% of controls (p<0.05).[27]

Small European studies assessed preventative strategies in lower doses of anthracyclines (average 240 mg/m²), but demonstrated that left ventricular ejection fraction (LVEF) measured by cardiac MRI declined less in patients treated with candesartan[28]; similar results are seen in those treated with enalapril and carvedilol.[29] Subsequent 2-year follow-up of the PRADA Study suggested only a small decline in LVEF in patients receiving lower dose anthracyclines, and no significant between group differences.[30]

ICOS-ONE randomised patients to enalapril or 'troponin-triggered' enalapril and found no difference between the two strategies.[31] Median dose of anthracyclines was low (180 mg/m²) and the most enalapril was typically prescribed at 2.5 mg two times per day.[31] The rate of cardiotoxicity at 3 years was low.[31] [32]

A UK based clinical trial, Cardiac CARE (ISRCTN24439460), is testing a different hypothesis to PROACT; patients with breast cancer or NHL will have troponin-guided randomisation to treatment with beta-blockers and ACEI or usual care. Cardiac CARE will provide complimentary information to PROACT on its completion.[33]

Troponin T is a sensitive marker of early cardiac cell death and as such there is the potential for troponin release to be 'turned off' if enalapril is effective. Audit data from 36 patients and 143 samples showed a positive troponin T in 47% of patients who received>300 mg/m² of anthracycline in a 6-cycle regimen. Troponin T was chosen as the primary outcome measure on this basis and was endorsed by our patient groups. It is known from previous research that troponin correlates with subsequent changes in LV function, and importantly that a negative troponin during and at 1 month post chemotherapy essentially excludes significant cardiotoxicity.[34] [35]

There are currently no definitive trials of ACEI in the prevention of anthracycline cardiotoxicity in patients receiving the highest contemporary doses of chemotherapy. PROACT will determine the effectiveness of enalapril in preventing cardiotoxicity in patients receiving high-dose anthracycline-based chemotherapy for breast cancer and NHL. Results will also inform practice for other cancer types. Findings will directly inform clinical practice in confirming whether enalapril should be given routinely to patients receiving anthracycline-based chemotherapy for breast cancer and NHL.

Increasing breast cancer and NHL survival, the frequency and impact of cardiotoxicity and the potential for a simple, safe and cheap preventative treatment makes PROACT highly important for patients and the NHS.

## METHODS AND ANALYSIS
### Study design
PROACT is a prospective, randomised, open-label, blinded end-point, superiority trial in patients undergoing anthracycline-based chemotherapy for breast cancer or NHL. Patients will be randomised to receive enalapril (intervention) or control (standard care).

The trial will answer the question 'can troponin release be prevented by enalapril in patients undergoing high-dose anthracycline chemotherapy treatment for cancer?'

### Setting
Patients due to receive high-dose anthracycline-based chemotherapy for their breast cancer or NHL at participating NHS Trusts will be offered recruitment to the trial. All sites can accommodate the needs of this trial including research nurse support, facilities for trial interventions and assessments, and British Society of Echocardiography (BSE) accredited echocardiographers, advanced trainee

or consultant cardiologists, to carry out echocardiograms in accordance with the trial protocol. The trial began recruiting in September 2017, and is due to report in late 2023.

## Eligibility criteria
### Inclusion criteria
► Adult patients with histopathologically confirmed breast carcinoma who have received surgery for their breast cancer (Patients with HER2+ breast cancer are eligible for inclusion).
► Planned to receive six cycles of EC 90 (total planned dose 540 mg/m$^2$ epirubicin) or FEC 75 (total planned dose 450 mg/m$^2$ epirubicin) adjuvant chemotherapy regimen.

OR
► Adult patients with histopathologically confirmed NHL planned to receive six cycles of R-CHOP or CHOP (total planned dose 300 mg/m$^2$ doxorubicin) chemotherapy. (Patients who will receive an alternative anti-CD20 monoclonal antibody are eligible (for exampleeg, O-CHOP), as long as the total planned doxorubicin dose is ≥300 mg/m2 over 6six cycles.)

AND
► Written informed consent.

### Exclusion criteria
► Positive baseline cardiac troponin T (≥14 ng/L).
► Known contraindication to ACEI, for example, renal artery stenosis, severe aortic stenosis.
► Are taking or have a previous intolerance to ACEI (eg, angioedema).
► Patient already taking other agents acting on the renin–angiotensin–aldosterone system, for example, Aliskiren, angiotensin receptor blockers, Entresto (sacubitril/valsartan), spironolactone, eplerenone.
► LVEF <50%.
► Estimated GFR<30 mL/min/1.73 m$^2$ at baseline.
► Hyperkalaemia defined as serum potassium≥5.5 mmol/L.
► Symptomatic hypotension or systolic blood pressure (SBP)<100 mm Hg.
► Poorly controlled hypertension (BP>160/100 mm Hg or ambulatory BP of 150/95 mmHg).
► Previous myocardial infarction.
► Known metastatic breast cancer.
► Previous exposure to anthracycline chemotherapy.
► Are pregnant or breast feeding.
► Previous trastuzumab treatment or planned trastuzumab treatment within 4 weeks following anthracycline chemotherapy.
► For patients of childbearing potential: refusal to use adequate contraception throughout trial. (Female patients between the ages of 18 and 50 will receive a pregnancy test at baseline).
► Any other invasive cancer diagnosed and treated in the past 5 years.

► Symptomatic or severe asymptomatic radiation-induced cardiac disease.
► Participation in other interventional medicinal trials in the past 6 months.
► Judgement by the investigator that the patient has a prognosis of <1 year or is unlikely to complete six cycles of chemotherapy.
► Judgement by the investigator that the patient is high risk for tumour lysis syndrome (applicable only to NHL patients).
► Judgement by the investigator that the patient should not participate in the study, (eg, if the patient is unlikely to comply with study procedures, restrictions and requirements).

## Randomisation
Randomisation will use a minimisation scheme, which adjusts for baseline factors; the minimisation scheme will account for the planned six cycle chemotherapy regimen (EC 90 or FEC 75 or R-CHOP), and, in breast cancer HER2 status (positive or negative). Patients will be randomised 1:1 to either standard care plus enalapril, or standard care only by members of the research team at each centre using a 24-hour, central, secure, web-based randomisation system with concealed allocation (procured from Sealed Envelope).

## Trial intervention
### Standard care
All patients in both arms will be planned to receive six cycles of chemotherapy as part of standard care.

There is no placebo in the standard care alone arm.

The following regimens are permitted within the trial:

Breast cancer regimens:
► EC 90; epirubicin 90 mg/m$^2$, cyclophosphamide 600 mg/m$^2$.
► FEC 75; fluorouracil 600 mg/m$^2$, epirubicin 75 mg/m$^2$, cyclophosphamide 600 mg/m$^2$.

Non-Hodgkin's lymphoma regimens:
► R-CHOP (Patients who receive an alternative anti-CD20 monoclonal antibody are eligible (eg, O-CHOP), as long as the total planned doxorubicin dose is ≥300 mg/m$^2$ over six cycles.) or CHOP; (rituximab 375 mg/m$^2$), cyclophosphamide 750 mg/m$^2$, doxorubicin 50 mg/m$^2$, vincristine 1.4 mg/m$^2$ (max 2 mg), prednisolone 40 mg/m$^2$ (for 5 days).

### Intervention arm
Patients randomised to the intervention arm will receive enalapril in addition to standard care. Patients will commence on a 2.5 mg two times per day dose at least 2 days prior to their first dose of anthracycline chemotherapy, with the aim to up-titrate the dose to 5 mg two times per day, then 10 mg two times per day over a maximum of three dose evaluation assessments prior to cycle 2. Chemotherapy will not be delayed by taking part in the trial. Some clinicians/centres routinely withhold ACEI on the morning of rituximab therapy; this is allowed per protocol.

Dose evaluation assessment visits will take place between 2 and 7 days after the start of each dose level. If the patient has systolic BP≥100 mm Hg, normal serum potassium (potassium<5.5 mmol/L) and stable renal function (serum creatinine<30 µmol/L), the dose will be increased to the next dose level. If clinical opinion is that the patient is unlikely to tolerate the higher dose, the dose will remain the same. If the patient has a ≥30 µmol/L increase in serum creatinine levels since the last assessment compared with baseline, new hyperkalaemia (potassium≥5.5 mmol/L), or symptomatic hypotension (SBP<100 mm Hg) then enalapril will be permanently discontinued. Patients will remain on the maximum enalapril dose reached for the duration of their chemotherapy, and until 3 weeks following their last dose of anthracycline. Temporary halts of up to 14 days will be allowed, with reintroduction and dose reductions at the discretion of the Investigator. Extensive patient and public involvement work at the point of trial design, and financial cost, led to a decision not to include a placebo in the control arm.

## Blinding
PROACT is an open-label trial. The trial employs a prospective randomised blinded endpoint design; analysis of troponin T and troponin I will be completed by laboratory staff who are blinded to the patients' trial allocation, as detailed in the trial PROACT laboratory manual. The trial management team, statistics and clinical teams will remain blinded to the troponin results until the end of the trial. The data manager, and an unblind monitor separate to the trial team, will have access to the troponin results for the purposes of monitoring and data cleaning.

All echocardiograms will be sent to an independent Core Laboratory for assessment by a BSE accredited echocardiographer/or an advanced trainee or consultant cardiologist blind to the intervention.

## Outcomes
### Primary outcome
The primary outcome is cardiotoxicity measured as presence (≥14 ng/L) of cardiac troponin T release at any time during anthracycline treatment, and 1 month after the last dose of anthracycline.

### Secondary outcomes
► Cardiac function will be assessed by echocardiogram, including global longitudinal strain (GLS), and measurements of LVEF, at baseline and following completion of chemotherapy.
► Cardiotoxicity will be measured as cardiac troponin I release during chemotherapy and at 1 month after the last dose of anthracycline.
► Adherence to enalapril will be measured according to patient diaries throughout the trial.
► All adverse events and adverse reactions including those that are serious and unexpected will be recorded from the day of randomisation until the last visit or until withdrawal; adverse events considered related to enalapril will be followed until resolution, a stable outcome or death.
► Anxiety or distress related to trial participation will be measured at the last study visit.
► Cancer and chemotherapy outcomes will be characterised in the population.

## Protocol changes
### Breast cancer management
Shortly after the trial opened to recruitment, changes in the clinical management of patients with breast cancer reduced the number of patients potentially eligible patients. The first was the widespread adoption of genomic testing to calculate the Oncotype Dx recurrence score, which reduced the number of patients recommended for adjuvant chemotherapy by approximately 30% in our recruiting hospitals.

In addition, in July 2018 NICE published a clinical guideline on the management of breast cancer, which included recommending a taxane to be routinely offered alongside an anthracycline in adjuvant chemotherapy (FEC-T chemotherapy); in the development of PROACT, we identified patients receiving FEC-T as being at low risk of cardiotoxicity. It was decided to include NHL patients in the trial to increase the number of potentially eligible patients.

### COVID-19 impact
PROACT clinical trial delivery has been significantly affected by the COVID-19 pandemic. The trial was paused to recruitment in March 2020, and while it has reopened to recruitment, ongoing impact on non-COVID-19 research in the NHS, and in particular recruitment to Oncology clinical trials, is well documented. National Institute for Health Research have agreed two funded extensions to enable the study to complete.

### Sample size
In light of the unprecedented challenges, the trial team, in agreement with funder, Trial Steering Committee (TSC) and Independent Data Monitoring and Ethics Committee (IDMEC) agreed to recalculate the sample size by changing from 90% to 80% power, using the same assumptions as in the original sample size calculation.

Assuming alpha of 5% and 80% power, 106 patients are needed to detect a reduction in the proportion of patients with cardiac troponin T present from 47% to 20% using a two-sided Fisher's exact test; additional recruitment to account for attrition is planned. Observational data was used to provide us with the estimate of troponin T elevation in the standard care arm; there are other potential causes of an elevated troponin, such as infection, during chemotherapy and clinical consensus indicated that a rate of 20% in elevated troponin would fully account these. There is also a consensus within the clinical community that, a large effect size will be necessary to convince the clinical community to change the pathway of care for

these patients. The original sample size at 90% power was 140 patients, inflated to 170 patients to account for attrition.

## Trial procedures

Patients due to undergo 6 cycles of anthracycline based chemotherapy for breast cancer or NHL at participating centres are identified by the clinical research team and approached about the trial, including given an information sheet and consent form. After discussion, consent is sought, baseline assessments performed and eligibility checked and confirmed. Eligible patients are randomised, and their general practitioners informed. A full schedule of events is detailed in table 1, and a participant flow chart provided in figure 1. Patients have assessments at baseline, dose evaluation visits (up to three for those receiving the intervention) within 72 hours prior to day 1 of each chemotherapy cycle, and at trial completion (4 weeks following the last dose of anthracycline). A separately funded study is allowing patients to be followed for one year, and all patients are asked to consent to longer term follow-up.

Baseline activities include consent, medical history, height/weight, New York Heart Association (NYHA) class, BP check, bloods, eGFR and eligibility check.

## Troponin T and troponin I

Blood sampling for troponin T and troponin I will be performed at baseline, within 72 hours prior to chemotherapy (cycle 2 onwards), and trial completion (30 days after the final anthracycline dose). Troponin T samples will provide the primary outcome data. Samples will be processed and sent to a core laboratory for central analysis; the central laboratory will be blind to treatment allocation.

## Echocardiogram

Cardiac function is assessed via transthoracic echocardiography (TTE) at baseline, 4 weeks after the last anthracycline dose and 1 year after the date of final chemotherapy. All echocardiograms will be assessed by a core laboratory who will be blind to participant treatment. Local reporting of TTEs will be as per local hospital practice; the data reported by the core laboratory (including GLS and LVEF) will be analysed as part of the secondary outcomes for the trial.

## Trial anxiety or distress participation assessment

At the start of cycle 4 and at the end of study visit (1 month after the last chemotherapy treatment), patients will be asked to complete a short questionnaire to understand if taking part in the trial has resulted in any anxiety or distress. This outcome was requested by the funding panel and will help to understand the impact on patients of a change to standard care in the event that PROACT demonstrates the effectiveness of enalapril in this setting.

## Statistical analysis

Data cleaning and analysis will be provided by staff within Newcastle Clinical Trials Unit and Durham University. Primary analysis will follow intention to treat principles with patient data analysed according to randomisation and irrespective of intervention received; other analysis groups such as perprotocol may be considered subsequently. Every effort will be made to retain and include all patients who are part of the trial.

A full statistical analysis plan will be developed for the outcome measures and agreed with the IDMEC and chief investigator prior to any analysis being undertaken.

Outcome data will be analysed at the end of the main study, no interim analysis is planned. Follow-up data will be analysed and reported separately.

The primary analysis of presence or absence of cardiac troponin T will be assessed using logistic regression and accounting for minimisation factors. Analysis of the secondary endpoints will be dependent on the nature of the specific endpoint and data structure; GLS and LVEF will be analysed as a change from baseline using regression model to compare intervention and control groups.

Cardiac troponin I data will also be analysed using logistic regression and account for minimisation factors. Changes in cardiac troponin T and I will also be analysed as a continuous variable. Additional analysis of the primary endpoint will be performed using logistic regression to account for baseline factors (regimen and HER2 status). Adverse events data will be analysed using cross-tabulation.

Sensitivity analysis will also be performed for adherence to the protocol and compare breast cancer and non-breast cancer patients.

## Trial conduct and governance

The trial management group is responsible for the day to day management of the trial, overseeing all aspects to ensure that the protocol is adhered to and taking appropriate actions to ensure patient safety and data integrity. The IDMEC review trial outcomes (including adverse events and serious adverse events), provide advice on the ongoing conduct and safety of the trial and report recommendations to the TSC. The TSC, where independent members are the majority (including patient representatives), provides overall supervision, monitors progress and conduct and advises on the trial. The IDMEC and TSC will meet every 6 months.

## Patient and public involvement

Initial trial ideas were discussed with patients who had recently finished chemotherapy treatment for breast cancer in the Maggie's Centre in Newcastle in 2015 and at a breast cancer support group in North Yorkshire and again with patients from the Maggie's Centre; trial ideas were developed between each discussion based on their feedback prior to submission of the application.

The trial also has two patient representatives who sit on the TSC; one patient previously treated for breast cancer,

**Table 1** PROACT schedule of events

| | Baseline (up to 6 weeks prior to randomisation) | Day 1 (at least 2 days prior to cycle 1 day 1) | Dose evaluation visit 1 | Dose evaluation visit 2 | Dose evaluation visit 3 | Cycle 1, day 1 | Cycle 2, day 1 | Cycle 3, day 1 | Cycle 4, day 1 | Cycle 5, day 1 | Cycle 6, day 1 | Trial completion visit | 1-year Follow-up visit |
|---|---|---|---|---|---|---|---|---|---|---|---|---|---|
| Consent | X | | | | | | | | | | | | |
| Demographics | X | | | | | | | | | | | | |
| Medical history | X | | | | | | | | | | | | X |
| Cancer history | X | | | | | | | | | | | | X |
| Performance status | X | | | | | | | | | | | | X |
| New York Heart Association (NYHA) class | X | | | | | | | | | | | | X |
| Concomitant medications | X | X | X | X | X | X | X | X | X | X | X | X | X |
| Physical assessment (height and weight) | X | | | | | | | | | | | | X |
| Blood pressure (BP) | X | | X | X | X | X | X | X | X | X | X | X | X |
| Troponin T (baseline sample) | X | | | | | | | | | | | | |
| Troponin T (post baseline) | | | | | | X | X | X | X | X | | X | |
| Troponin I | X | | | | | X | X | X | X | X | | X | |
| Urea and Electrolytes (U+Es) | X | | X | X | X | X | X | X | X | X | | X | |
| estimated Glomerular Filtration Rate (eGFR) | X | | | | | | | | | | | | |
| Pregnancy test | X | | | | | | | | | | | | |
| Echocardiogram | X | | | | | | | | | | | X | X |
| Eligibility check | X | | | | | | | | | | | | |
| Randomisation | X | | | | | | | | | | | | |
| Enalapril (intervention group only) | | X | X | X | X | X | X | X | X | X | | | |
| Adherence to enalapril (intervention group only) | | X | X | X | X | X | X | X | X | X | | X | |
| Standard care chemotherapy (all patients) | | | | | | X | X | X | X | X | | | |
| Adverse Events | | X | X | X | X | X | X | X | X | X | | X | |
| Buccal swab for future research | X | | | | | | | | | | | | |
| Blood sample for future research | X | | | | | | X | | X | | | X | |
| Acceptability of trial interventions assessment | | | | | | | | | X | | | X | |

NYHA, New York Heart Association; PROACT, Preventing cardiotoxicity in patients with breast cancer and lymphoma trial.

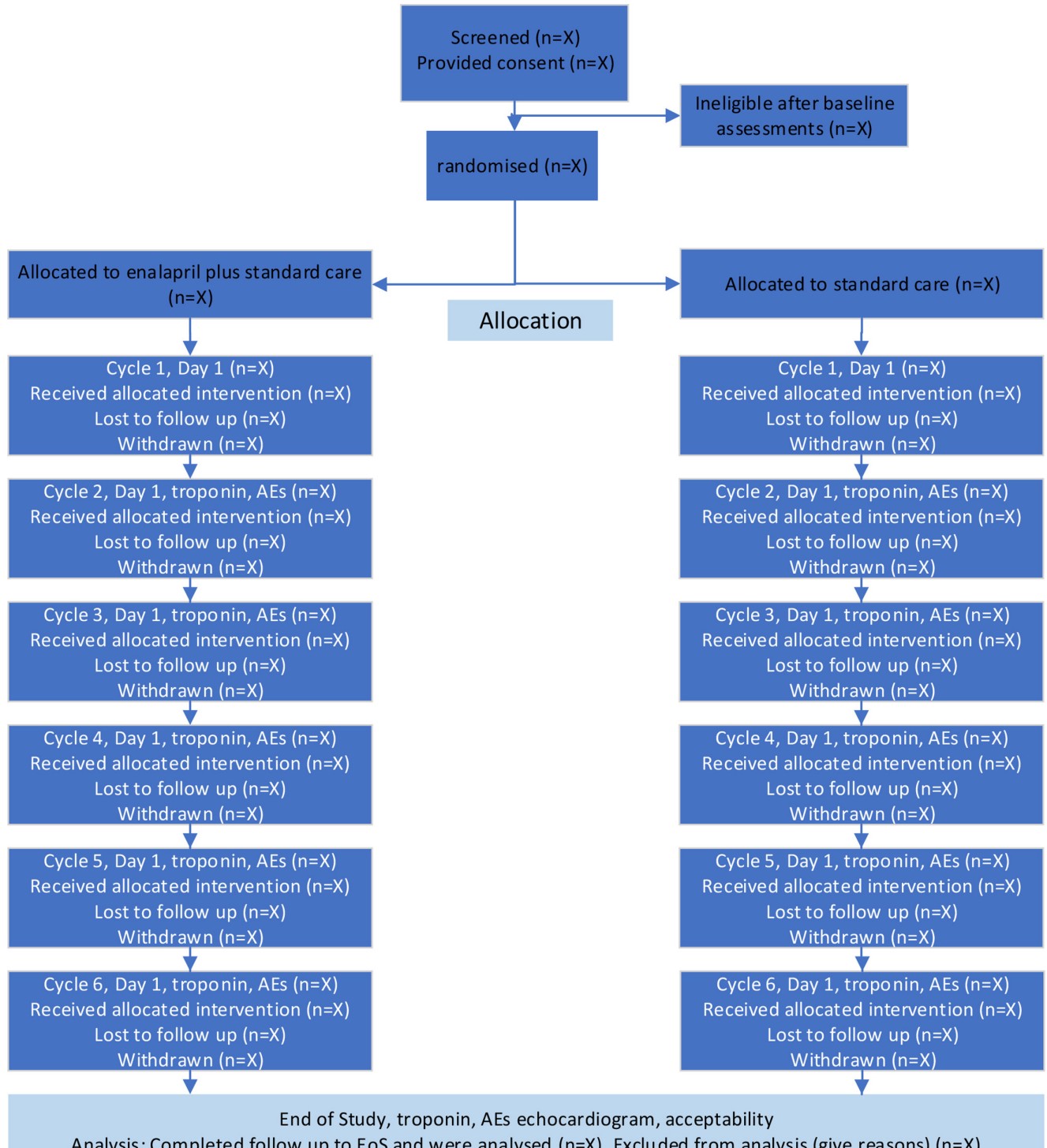

**Figure 1** PROACT trial flow diagram.

the other previously treated for NHL. Neither of these patients are trial participants, nor are they employed by any organisation directly involved in the trial conduct.

### Ethics and dissemination

All clinical research raises ethical issues, and the trial team carefully thought through potential issues and aimed to address these in the trial design. This study does not include patients who will lack capacity, nor minors. This trial is not based in an emergency setting. The trial is conducted in accordance with the protocol (current V.8.0, dated on 3 March 2022), good clinical practice, the favourable ethical opinion (from West Midlands—Edgbaston REC, Ref: 17/WM/0248) and the MHRA notice of no objection. All patients provide written informed consent prior to participation. Enalapril has been widely used for over thirty years, is well tolerated without significant side effects.

If this research demonstrates that enalapril is effective in preventing heart damage in patients receiving anthracycline chemotherapy, this will have a significant impact not just for patients with breast cancer and NHL, but for both adult and paediatric patients with other types of cancer treated with chemotherapy regimens containing anthracyclines. In addition to reduced morbidity and mortality for patients, enalapril will provide significant savings for the NHS.

We will publish the findings in peer-reviewed journals, and disseminate results to patients, and the international clinical community.

**Author affiliations**
[1]Newcastle Clinical Trials Unit, Newcastle University, Newcastle upon Tyne, UK
[2]South Tees Hospitals NHS Foundation Trust, Academic Cardiovascular Unit, South Tees, Middlesbrough, UK
[3]Population Health Sciences Institute, Newcastle University, Newcastle upon Tyne, UK
[4]Department of Cardiology, Newcastle Upon Tyne Hospitals NHS Foundation Trust, Newcastle Upon Tyne, UK
[5]Department of Anthropology, Durham University, Durham, UK
[6]Mathematical Sciences, University of Durham, Durham, UK
[7]Department of Haematology, South Tees Hospitals NHS Foundation Trust, Middlesborough, UK
[8]Department of Oncology, South Tees Hospitals NHS Foundation Trust, Middlesbrough, UK
[9]Department of Cardiology, South Tees Hospitals NHS Foundation Trust, Middlesbrough, UK
[10]Outreach Research & Innovation Group, Didsbury, UK
[11]Department of Oncology, County Durham and Darlington NHS Foundation Trust, Darlington, UK
[12]Department of Cardiology, North Tees and Hartlepool NHS Foundation Trust, Hartlepool, UK
[13]Department of Oncology, Newcastle Upon Tyne Hospitals NHS Foundation Trust, Newcastle Upon Tyne, UK

**Acknowledgements** We are grateful to members of the Independent Data Monitoring and Ethics Committee, and Trial Steering Committee for their support of PROACT. We would like to acknowledge the considerable work being undertaken by principal investigators and members of the site teams in support of this trial, and the PROACT trial team at Newcastle Clinical Trials Unit.

**Contributors** DA and CP conceived the idea for the trial. DA, HCH, RHM, ASK, CP, MV, NC, JG, JM, MS, AW, SH, HO, AH and RG codesigned the trial, secured funding from the National Institute for Health Research for Patient Benefit and wrote the full trial protocol with substantial input from LC. DA is the chief investigator. RHM and HCH provided methodological input and oversee Newcastle Clinical Trials Unit activity. EO led the statistical aspects and analysis, overseeing NA. SV undertook echocardiogram review and reporting at the core echocardiogram laboratory. This paper was drafted from the approved version of the protocol; all authors commented and amended drafts of the paper and approved the final manuscript.

**Funding** This work is supported by the National Institute for Health Research for Patient Benefit Programme (Ref: PB-PG-0815-20061). The views expressed are those of the authors and not necessarily those of the NIHR or the Department of Health and Social Care.The PROACT Trial is sponsored by South Tees Hospitals NHS Foundation Trust.

**Competing interests** ASK's contribution was during his employment by Durham University. He currently works for UCB Biopharma, UK. AW currently works for Outreach Research & Innovation Group and was employed by The Christie NHS Foundation Trust, Manchester, when the grant was awarded. DA has previously received speaker fees from AstraZeneca, Pfizer and Philips/Volcano. None were directly relevant to PROACT.

**Patient and public involvement** Patients and/or the public were involved in the design, or conduct, or reporting, or dissemination plans of this research. Refer to the Methods and analysis section for further details.

**Patient consent for publication** Not applicable.

**Provenance and peer review** Not commissioned; externally peer reviewed.

**ORCID iDs**
Rebecca H Maier http://orcid.org/0000-0002-7350-3288
Adetayo S Kasim http://orcid.org/0000-0002-0411-3059
Nasima Akhter http://orcid.org/0000-0002-5424-1593
Jamie Maddox http://orcid.org/0000-0002-7995-7934
Nicola Cresti http://orcid.org/0000-0002-0495-4472
Helen C Hancock http://orcid.org/0000-0002-1494-8551
David Austin http://orcid.org/0000-0003-4606-1055

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
