## [Reviewer comments · BMJ Open]

ARTICLE DETAILS

TITLE (PROVISIONAL)	Preventing Cardiotoxicity in Patients with Breast Cancer and Lymphoma: protocol for a multi-centre randomised controlled trial (PROACT)
AUTHORS	Maier, Rebecca; Plummer, Chris; Kasim, Adetayo; Akhter, Nasima; Ogundimu, Emmanuel; Maddox, Jamie; Graham, Janine; Stewart, Michael; Wardley, Andrew; Haney, Sophie; Vahabi, Sharareh; Oxenham, Helen; Humphreys, Alison; Cresti, Nicola; Verrill, Mark; Graham, Richard; Chang, Lisa; Hancock, Helen; Austin, David

VERSION 1 – REVIEW

REVIEWER	Peter Henriksen The University of Edinburgh Centre for Cardiovascular Science
REVIEW RETURNED	25-Sep-2022

GENERAL COMMENTS	The PROACT study is examining whether enalapril can prevent increased cardiac troponin T concentrations (above the 99th centile URL) in breast cancer and non-Hodgkin lymphoma patients receiving anthracycline chemotherapy. The study question is relevant. The recently released European Cardio-oncology guidelines recommend consideration of prescription of ACEi in PROACT-type patients receiving high dose anthracycline regimens. They also recommend cTn monitoring and define mild asymptomatic chemotherapy related cardiac dysfunction as elevation in cTn concentration in presence of preserved (>50%) LV ejection fraction. It could be argued that by inference, cTn has become a therapeutic target. The clinical trial evidence supporting these recommendations is weak. The manuscript could be improved with some further clarification. 1. Randomisation and minimisation Prognostic factors for developing a cTn concentration above the 14 ng/L URL include sex (men have higher concentrations), baseline cTn concentration (patients with baseline cTn of 10-13 ng/L more at risk than < 5ng/L) and cumulative anthracycline dose. The plan is to minimise for anthracycline dose and HER2 status (indicating the need for subsequent anti-HER2 therapy). Anti-HER2 therapies will presumably be delivered after completion of the primary end-point. Anti-HER2 treatment has definite effects on LV ejection fraction but effects on cTn concentration are less clear. The authors should explain the choice of minimisation and address the concern that imbalances of sex and baseline cTn concentrations between groups could influence the outcome. 2. cTn concentrations and LVEF decline/cardiotoxicity The authors state on page 6, ln 46 that 'a negative troponin
---

	essentially excludes significant cardiotoxicity' The references are for older studies conducted with contemporary (non high sensitivity assays). The detection limit for these contemporary assays was for cTn concentrations up to 10x higher than 99th centile URL for high sensitivity assays. Because hs-cTn is being used as a surrogate for LVEF decline and cardiotoxicity, the authors should discuss what is known about high sensitivity cTn monitoring and subsequent LVEF decline for anthracycline treated patients- there has been a recent meta-analysis looking at LVEF change including hs-cTn assays. Could it be that a hs-cTnT of 30 ng/L is 'elevated' but still 'safe'? 3. Non-anthracycline related causes of cTn elevation On page 9 the authors describe how 'clinical consensus indicated..that a rate of 20% would fully account for this..'. Such a significant and frequent effect on the primary outcome by undefined factors is concerning. The authors should explain how they identified this figure and the potential for variation between groups to influence the outcome. Will they attempt to record and analyse infection episodes between groups given that these are listed as a cause? 4. Outcomes The study is powered around the the primary outcome of cardiac troponin T concentration increase above 14 ng/L. Echocardiographic measures of LVEF decline and GLS fall are secondary endpoints that are clinically relevant. The authors should describe in the protocol paper whether they are powered to detect change in these measures and comment on whether they feel a positive treatment impact on cTnT but neutral effect of LVEF would change clinical practice. 5. ESC Guidelines 2022 The authors should consider referring to this guideline that was released recently but sets the context and relevance of the PROACT Trial
--	---

REVIEWER	Edimar Bocchi Universidade de Sao Paulo Instituto do Coracao, Heart Failure Clinics
REVIEW RETURNED	26-Sep-2022

GENERAL COMMENTS	In summary: The authors submitted the PROACT study that is is a prospective, randomised, open-label, blinded end-point, superiority trial which will recruit adult patients being treated for breast cancer and NHL at NHS hospitals throughout England. The trial aims to recruit 106 participants, who will be randomised to standard care (high dose anthracycline based chemotherapy) plus enalapril (intervention), or standard care alone (control).PROACT will determine the effectiveness of enalapril in preventing cardiotoxicity in patients receiving high-dose anthracycline-based chemotherapy for breast cancer and NHL. This work is supported by the National Institute for Health Research (NIHR) Research for Patient Benefit (RfPB) Programme.
---

VERSION 1 – AUTHOR RESPONSE

Reviewer: 1

Dr. Peter Henriksen , The University of Edinburgh Centre for Cardiovascular Science

Comments to the Author:

The PROACT study is examining whether enalapril can prevent increased cardiac troponin T concentrations (above the 99th centile URL) in breast cancer and non-Hodgkin lymphoma patients receiving anthracycline chemotherapy. The study question is relevant. The recently released European Cardio-oncology guidelines recommend consideration of prescription of ACEi in PROACT-type patients receiving high dose anthracycline regimens. They also recommend cTn monitoring and define mild asymptomatic chemotherapy related cardiac dysfunction as elevation in cTn concentration in presence of preserved (>50%) LV ejection fraction. It could be argued that by inference, cTn has become a therapeutic target. The clinical trial evidence supporting these recommendations is weak.

The manuscript could be improved with some further clarification.

1. Randomisation and minimisation

Prognostic factors for developing a cTn concentration above the 14 ng/L URL include sex (men have higher concentrations), baseline cTn concentration (patients with baseline cTn of 10-13 ng/L more at risk than < 5ng/L) and cumulative anthracycline dose.

The plan is to minimise for anthracycline dose and HER2 status (indicating the need for subsequent anti-HER2 therapy). Anti-HER2 therapies will presumably be delivered after completion of the primary end-point. Anti-HER2 treatment has definite effects on LV ejection fraction but effects on cTn concentration are less clear. The authors should explain the choice of minimisation and address the concern that imbalances of sex and baseline cTn concentrations between groups could influence the outcome.

Thank you. We agree with these comments and plan to include an analysis of cTnT as a continuous measure to understand the impact of different values on outcome. We also plan to include an analysis to understand the impact of sex -specific troponin thresholds and will add baseline (below the 99th URL) troponin to our sensitivity analysis in the Statistical Analysis Plan.

The primary end point assessment is prior to Herceptin treatment, but we wished to preserve balance for this factor as longer term clinical and echo follow up is planned.

We chose to balance for anthracycline dose as a key determinant of cardiotoxicity risk. We have included a number of different high dose anthracycline regimens in this trial, reflecting clinical practice and wanted to ensure that the overall planned dose of anthracycline was similar in each arm of the trial.

2. cTn concentrations and LVEF decline/cardiotoxicity

The authors state on page 6, In 46 that 'a negative troponin essentially excludes significant cardiotoxicity' The references are for older studies conducted with contemporary (non high sensitivity assays). The detection limit for these contemporary assays was for cTn concentrations up to 10x higher than 99th centile URL for high sensitivity assays. Because hs-cTn is being used as a surrogate for LVEF decline and cardiotoxicity, the authors should discuss what is known about high sensitivity cTn monitoring and subsequent LVEF decline for anthracycline treated patients- there has been a recent metanalysis looking at LVEF change including hs-cTn assays. Could it be that a hs-cTnT of 30 ng/L is 'elevated' but still 'safe'?

Dr Henriksen makes several salient points regarding cTn in this field, particularly our reliance on more historical data, and older troponin assays to inform our trial design. As noted, the ESC/ICOS published inaugural guidelines that defined "mild cardiotoxicity" as an elevation from baseline in cardiac biomarkers. The PROACT trial primary end point is therefore consistent with this new definition, although we agree that it remains to be seen whether small elevations in hs-cTn are

meaningful. We think it remains highly likely that a negative hs-cTn during and after anthracycline treatment will exclude cardiotoxicity, and therefore this central tenet of the study design remains valid.

We have excluded all patients from the study with a positive hs-cTnT at baseline, recognising that we do not know what the impact of mildly raised troponin is on LVEF. We will be able to begin to understand if there are thresholds of elevated troponin that are 'safe' or otherwise, through our secondary and exploratory analyses of troponin as a continuous measure, and at different thresholds, in relation to echocardiographic findings.

3. Non-anthracycline related causes of cTn elevation

On page 9 the authors describe how 'clinical consensus indicated..that a rate of 20% would fully account for this..'. Such a significant and frequent effect on the primary outcome by undefined factors is concerning. The authors should explain how they identified this figure and the potential for variation between groups to influence the outcome. Will they attempt to record and analyse infection episodes between groups given that these are listed as a cause?

We discussed this at length with several clinicians, with infection felt to be the most likely cause for an elevated troponin during treatment. The figure of 20% was therefore a clinical consensus, rather than based on specific data. We will be able to present data on infection rates in both groups when we publish our findings to help further explain any differences seen.

4. Outcomes

The study is powered around the the primary outcome of cardiac troponin T concentration increase above 14 ng/L. Echocardiographic measures of LVEF decline and GLS fall are secondary endpoints that are clinically relevant. The authors should describe in the protocol paper whether they are powered to detect change in these measures and comment on whether they feel a positive treatment impact on cTnT but neutral effect of LVEF would change clinical practice.

We have not powered the study to detect change in the measures of LVEF decline and GLS fall. Given the guideline below, which advocates the use of hs-troponin as a biomarker of cardio-toxicity we believe that a positive impact on cTnT in PROACT could indeed change practice. We are undertaking a follow-on study looking at echocardiographic measures beyond one year and believe that once published, this would help to understand the likely impact of findings if a neutral effect on LVEF is seen within the main trial.

5. ESC Guidelines 2022

The authors should consider referring to this guideline that was released recently but sets the context and relevance of the PROACT Trial

Thank you, these guidelines were released following submission of our paper to BMJOpen. We have now included them as a reference.

Reviewer: 2

Dr. Edimar Bocchi, Universidade de Sao Paulo Instituto do Coracao

Comments to the Author:

In summary: The authors submitted the PROACT study that is is a prospective, randomised, open-label, blinded end-point, superiority trial which will recruit adult patients being treated for breast cancer

and NHL at NHS hospitals throughout England. The trial aims to recruit 106 participants, who will be randomised to standard care (high dose anthracycline based chemotherapy) plus enalapril (intervention), or standard care alone (control).PROACT will determine the effectiveness of enalapril in preventing cardiotoxicity in patients receiving high-dose anthracycline-based chemotherapy for breast cancer and NHL. This work is supported by the National Institute for Health Research (NIHR) Research for Patient Benefit (RfPB) Programme .

Many thanks for your comments